# Optimized Green Extraction of Polyphenols from *Cassia javanica* L. Petals for Their Application in Sunflower Oil: Anticancer and Antioxidant Properties

**DOI:** 10.3390/molecules27144329

**Published:** 2022-07-06

**Authors:** Mohamed Ibrahim Younis, Xiaofeng Ren, Azalldeen Kazal Alzubaidi, Khaled Fahmy Mahmoud, Ammar B. Altemimi, Francesco Cacciola, Husnain Raza, Anubhav Pratap-Singh, Tarek Gamal Abedelmaksoud

**Affiliations:** 1School of Food and Biological Engineering, Jiangsu University, Zhenjiang 212013, China; mohamedyounis2201@agr.cu.edu.eg (M.I.Y.); raza@ujs.edu.cn (H.R.); 2Food Science Department, Faculty of Agriculture, Cairo University, Giza 12613, Egypt; tareekgamal_88@agr.cu.edu.eg; 3College of Agriculture, University of Misan, Al-Amara 62001, Iraq; ez_aldeen@uomisan.edu.iq; 4Food Technology Department, National Research Center, Giza 12613, Egypt; kf.mahmoud@nrc.sci.eg; 5Scientific Research Center, Al-Ayen University, Nasiriyah 64001, Iraq; ammar.ramddan@uobasrah.edu.iq; 6Department of Food Science, College of Agriculture, University of Basrah, Basrah 61004, Iraq; 7Department of Biomedical, Dental, Morphological and Functional Imaging Sciences, University of Messina, Via Consolare Valeria, 98125 Messina, Italy; cacciolaf@unime.it; 8Institute for Advanced Study (IAS), Shenzhen University, No. 3688, Nanhai Avenue, Nanshan District, Shenzhen 518060, China; 9Food, Nutrition & Health Program, Faculty of Land and Food Systems, The University of British Columbia, Vancouver, BC V6T 1Z4, Canada

**Keywords:** *Cassia javanica*, SCF-CO_2_, nano-encapsulated, antioxidant, β-carotene/linoleic acid

## Abstract

The total phenolic content (TPC) from *Cassia javanica* L. petals were extracted using ethanolic solvent extraction at concentrations ranging from 0 to 90% and an SCF-CO_2_ co-solvent at various pressures. Ultrasound-assisted extraction parameters were optimized using response surface methodology (RSM). Antioxidant and anticancer properties of total phenols were assessed. An SCF-CO_2_ co-solvent extract was nano-encapsulated and applied to sunflower oil without the addition of an antioxidant. The results indicated that the best treatment for retaining TPC and total flavonoids content (TFC) was SCF-CO_2_ co-solvent followed by the ultrasound and ethanolic extraction procedures. Additionally, the best antioxidant activity by β-carotene/linoleic acid and DPPH free radical-scavenging test systems was observed by SCF-CO_2_ co-solvent then ultrasound and ethanolic extraction methods. SCF-CO_2_ co-solvent recorded the highest inhibition % for PC3 (76.20%) and MCF7 (98.70%) and the lowest IC_50_ value for PC3 (145 µ/mL) and MCF7 (96 µ/mL). It was discovered that fortifying sunflower oil with SCF-CO_2_ co-solvent nanoparticles had a beneficial effect on free fatty acids and peroxide levels. The SCF-CO_2_ method was finally found to be superior and could be used in large-scale processing.

## 1. Introduction

Natural compounds in plants for maintaining human health started to be identified a rather long time ago, though the identification of active compounds from this valuable source came to light in recent years. The potential adverse effects of plant-based products are very limited in addition to relatively inexpensive compared to chemical substances [1]. As a result, numerous researchers have focused on plants that contain biologically active compounds known as organic phytochemicals [2]. *Cassia javanica* is one of these plants. It is an ornamental evergreen plant that is found throughout the world. It is used to treat gastric pain, the common cold, malaria, measles, chickenpox, and constipation in China. For total soluble sugars, total soluble proteins, carotenoids, total free amino acids, lipids, and total phenol fresh *Cassia javanica* flower extract contained 37.83 mg/g, 52.75 mg/g, 0.731 mg/g, 6.130 mg/g, 13.33 mg/g, and 32.81 mg/g, respectively [3]. Additionally, the *Cassia javanica* flower contains alkaloids, cardiac glycosides, flavonoids, phenolic compounds, lobatannins, tannins, triterpenoids, volatile oils, and saponins. However, research on *Cassia javanica* is scarce to gain a better understanding of its properties, safety, and efficiency [3].

Traditionally, organic solvents were utilized to extract phytochemicals from their plant sources. These methods were initially beneficial, but as they evolved, some downsides became apparent [4]. The disadvantages of organic solvent extraction include a lack of selectivity for extracted components and difficulties with solvent separation from the extract. Solvent separation is an energy-intensive and sometimes dangerous procedure. Additionally, the evaporation of solvents is likely to result in the destruction of certain chemicals included in the extract [5]. In addition, solvents cannot be entirely extracted from post-extraction residues, resulting in solvent loss and restrictions on the use of post-extraction residues in feeds or spice combinations. The primary concern, however, is the presence of volatile organic contaminants in the finished product [4]. Numerous new technologies, both thermal and non-thermal, have been employed to extract phytochemicals from their sources. However, non-thermal (green) approaches were superior to thermal procedures, which retained more phytochemicals. Ultrasound and supercritical carbon dioxide are two of the more current environmentally friendly technologies (SCF-CO_2_). Ultrasound technology, which is employed in food processing in both liquid and solid media, is gaining substantial interest at the moment. Ultrasound has a number of effects on liquid systems, including cavitation, heating, dynamic agitation, shear stresses, and turbulence. It is critical in food technology, particularly in the processing, preservation, and extraction of food. At the moment, highly reproducible food processing may be finished in seconds utilizing ultrasonic, which reduces processing costs, simplifies manipulation, results in a purer final product, and consumes less time and energy than conventional techniques. Ultrasound treatment is also reported to significantly alter the conductivity and rheological properties of the sample and enhanced its solubility, specific surface area, and index of emulsifying activity [6,7].

Several research groups have recently begun to investigate supercritical fluids and their potential use as process solvents in a variety of applications. Due to the fact that SCF- CO_2_’s solubility can be controlled via temperature and pressure manipulation, it provides novel opportunities for selective extractions and fractionations [8,9]. CO_2_’s non-toxicity and non-flammability are critical for its use in the food business. According to the available literature, no investigations on the use of ultrasound and SCF-CO_2_ to extract bioactive components from *Cassia javanica* Flower Petals have been done. Thus, the purpose of this study is to extract phenolic compounds from *Cassia javanica* Petals Powder (CJDP) using a variety of extraction methods, including solvent extraction, ultrasound-assisted extraction, and SCF-CO_2_ extraction, and to investigate the activities of these extracted compounds before applying them to sunflower oil without added antioxidants.

## 2. Results and Discussion

### 2.1. Total Phenolic Content of Cassia javanica Petals Powder and Ethanol Concentration

Total phenolic content extraction from plant materials is influenced by phytochemical nature, sample composition, extraction method, chemical nature and polarity of solvents, sample particle size, solid–solvent ratio, and physical extraction conditions. The solvent is the most critical parameter when time and temperature are constant [10,11]. Therefore, in the current study, the effect of ethanol various concentrations (0–90%) on the total phenolic content (TPC) of *Cassia javanica* Petals Powder (CJPD) to obtain optimum concentration was demonstrated. From Figure 1, the TPC of CJPD extract was significantly greater at 60% ethanol (92.10 mg GAE/g dw) than at other ethanol concentrations.

According to the results of Figure 1, the authors concluded that the extraction yield of CJPD increases with increasing concentration of the extracting solvent (ethanol) used up to 60%. Both ethanol and water are widely recognized as safe solvents, and the use of hydroethanolic solutions to extract phenolic chemicals and decrease solvent costs has been documented in the scientific literature. In this regard, several studies were demonstrated that using 50%, 60%, or 75% (ethanol/water, *v*/*v*) ethanolic solutions as solvent extraction improves phenolic extraction from flaxseeds, guarana, and propolis extracts, respectively [12,13,14].

### 2.2. Optimization of Ultrasound Parameters for Assisted Extraction of TPC

The utilization of ultrasound-assisted TPC extraction of CJDP was conducted using the following parameters sonication time (χ^1^) and sonication power (χ^2^). The selected levels of sonication time (χ^1^) and sonication power (χ^2^) that resulted in the highest TPC were (5–15 min) and (100–400 watt) (Figure 2). These levels were used for response surface methodology (RSM) to evaluate their effect of them on the TPC values (mg/g) as a response as well as to optimize process parameters. The TPC of CJDP varied between 82.88 and 113.5 mg/g (Table 1). Table 1 shows the effect of ultrasound parameters on the TPC in the CJDP at a 95% confidence interval using ANOVA analysis.

The effect of ultrasound parameters (sonication time (χ^1^) and sonication power (χ^2^)) on the extraction yield of CJDP phenolic compounds showed that they were very effective in extracting these compounds with the increase in sonication time and sonication power (Table 1, Figure 2).

All responses exhibited a significant sum of squares and higher regression coefficients, indicating conformity with the variables specified by a second-order polynomial (Equation (1)). The results of fitting linear, interactive (2 FI), quadratic, and cubic regression models to the experimental data are presented in Table 1. The experimental data are best represented by the quadratic model, according to the information presented in Table 1. Except for the cubic model, which was aliased or confounded, the quadratic model had the highest and most significant values for the coefficient of determination (R^2^), the adjusted coefficient of determination (Adj-R^2^), and the expected coefficient of determination (Pred-R^2^), among other models (Table 2). Given below are the multiple regression equations generated between the various responses and process variables (Equation (1)):TPC = +90.52 + 4.99 χ^1^ + 7.89 χ^1^ + 0.16 χ^1^ χ^2^ − 0.14 χ^12^ + 6.72 χ^22^(1)

Appropriate model parameters (*p* < 0.05) are (χ^1^) and χ^1^ (sonication Time), χ^2^ (sonication power), χ^1^, χ^2^, χ^12^, χ^22^ are significant model terms (*p* < 0.05). Regression coefficients magnitude designated maximum positive effect ultrasound parameters, for which the TPC was 110.14 mg/g for the optimum conditions (Table 1).

As illustrated in Figure 2, the TPC increased as the sonication time (factor A) and power (factor B) increased (factor B). The surface plots of the model equation, disturbance, and 3D response demonstrate that factors A and B have a significant effect on TPC values. The model equation, perturbation, and three-dimensional response surface plots demonstrate that all factors have a significant effect on TPC values. The perturbation plot (Figure 2) depicted the relative impact of the operating parameters on the target response in the following order: TPC values: factor B > factor A.

The optimal conditions that were chosen are listed in Table 1. CJDP was prepared again under these conditions, and the experimental response values matched the predicted value using the optimized CJDP to extract the TPC model; thus, the fitted models were found to be suitable for predicting the responses (Table 3). Table 3 also shows the values of TPC and Total flavonoids content (TFC) for the optimum conditions of each treatment, which the TPC and TFC were 92.1 and 67.43 mg/g, 112.89 and 82.54 mg/g, 177.58 and 139.85 mg/g for Ethanol 60%, Sonication (15 min, 400 watt), and SCF-CO_2_ treatments, respectively.

### 2.3. Supercritical Fluid Carbon Dioxide (SCF-CO_2_) Assisted Extraction

Supercritical fluid CO_2_ is a non-polar solvent, although temperature and pressure variations can alter its polarity. Although flavonoids with a large molecular mass are insoluble in pure CO_2_, their solubility can be increased by adding a polar modifier or increasing the pressure [15].

The results in Figure 3 illustrate the TPC of extracts obtained from CJPD using SCF-CO_2_ and a co-solvent at a temperature of 40 °C. The effect of variation in pressure of SCF-CO_2_ (150–400 bar) on the phenolic content of CJPD was investigated. The TPC of the extracts obtained from CJPD using SCF-CO_2_ with co-solvent at 40 °C is shown in Figure 3. TPC concentrations ranged from 75.58 to 177.58 mg GAE/g. At 250 bar, the total phenolic content of the CJPD extract (177.58 mg GAE/g dw) was significantly higher than at other SCF-CO_2_ pressures (150, 200, 300, 350, and 400 bar). According to the results in Figure 3, the total phenolic content decreased to 22.92 mg/g as the pressure level increased. This reduction may be a result of the volatile and polar nature of the extracted target chemicals [16,17,18]. At 250 bar, it was observed that the effect of pressure on the density and solvation power of TPC. As a result of the low energy required to pressurize the system and the high phenolic yield, 250 bar and 40 °C were chosen as the optimal conditions for extracting TPC from CJDP.

### 2.4. The Effect of Different Extraction Methods on the Identification of the Phenolic Compounds of Cassia javanica Petals

Foods contain significant amounts of bioactive compounds (phenolic and flavonoids) that act as antioxidants and anticancer agents. The fractionation of phenolic compounds according to the optimal conditions for each treatment (Ethanol, ultrasound-assisted extraction, and SCF-CO_2_ co-solvent) during this study was conducted (Table 4, Appendix A). Table 4 showed that the concentration of identified phenolic compounds in SCF-CO_2_ contained the highest concentration followed by ultrasound and ethanolic treatments. The majority of the phenolic compounds noticed in CJPD were Pyrogallol, gallic acid, catechol, chlorogenic acid, *p*-hydroxybenzoic acid, syringic acid, ferulic acid, benzoic acid, vanillic acid, rutin, elagic acid, o-coumaric acid, cinnamic acid, Quercitin, Rosemarinic, Myricetin, and Kampherolin) as in Table 4. This increase in phenolic compounds, TPC and also TFC extraction for SCF-CO_2_, may be attributed to their unique tunable physical properties, such as changeable density, liquid/gas-like viscosity, and high diffusivity [19]. Meanwhile, the ultrasound treatment attributed to cavitation phenomena makes cells collapse, thus leading to more extraction for TPC, and TFC also compared to the ethanolic extraction method [20].

### 2.5. Cytotoxic Effect on Human Cell Lines for All Treated Samples

To assess the anticancer effect of the CJDP extract (high in phenolic compounds), two cancer cell lines (breast and prostate) were treated with ethanol, ultrasound, and SCF-CO_2_. The results were compared to those obtained using the standard drug, vinblastine sulphate. According to Figure 4B, the SCF-CO_2_ extract inhibited PC3 and MCF7 cells at the highest levels (76.2 and 98.7%, respectively). This is due to the fact that it contains a higher concentration of total phenolic compounds than other treatments. Additionally, there was a direct correlation between TPC concentration and cancer cell inhibition (breast and prostate cancer) (Figure 4A). Appendix A showed the relationship between TPC concentration and the inhibition of cancer cells (breast, and prostate cancer). Numerous studies have previously reported the anticancer effect of phenolic compounds. This effect was attributed primarily to the presence of antioxidant compounds (most notably TPC and TFC) [21]. Our results were similar to Wieczynska et al. [22], indicating a significant reduction in mitochondrial activity, which may have contributed to a decline in cellular viability. On the other hand, IC_50_ is described as the lethal concentration of the sample which causes the death of 50% of cells in 48 h (there is an inverse relationship between inhibition % and IC_50_), therefore, compared to the positive control (Doxorubicin, an anticancer drug for hematologic and solid tumors, was used as the positive control), the least values belonged to SCF-CO_2_ in the case of MCF7 (96 µg/mL), and PC3 (145 µg/mL) was the best followed by ultrasound (130 µg/mL) and solvent extract (165 µg/mL) for MCF7 cells and for PC3 cells; solvent extract (168 µg/mL) came first and then ultrasound (180 µg/mL) as shown in Figure 4B. This is attributed to the high level of Antioxidant % for SCF-CO_2_ treatment as presented in Figure 5A,B compared to other treatments (which is mainly attributed to phenolic compounds and flavonoid content as in Table 3).

### 2.6. β-Carotene/linoleic Acid and DPPH Assay of All Treated Samples

A complementary test system, namely the β-carotene/linoleic acid system, was used to screen the extracts obtained through the three extraction procedures for their potential antioxidant activity. In the overall activity test, SCF-CO_2_ was found to be superior to the other treatments. As a result, the antioxidant activity of the CJDP-derived SCF-CO_2_ extract is now considered. The results in Figure 5A demonstrate that the β-carotene/linoleic acid extract of *Cassia javanica* extracted using SCF-CO_2_ outperformed all other extracts in the β-carotene/linoleic acid system (95.85 ± 1.38%). The capability of DPPH radicals to undergo reduction was determined by a decrease in their absorbance at 517 nm. The decrease in DPPH radical absorbance is a result of hydrogen donation scavenging by the antioxidant. DPPH assay showed that the SCF-CO_2_ extract of CJDP had the highest scavenging with 65.7% followed by ultrasound with 46.8%, and the ethanolic extract with 41.2% as shown in Figure 5B. In addition, BHT values were 97% and 19.2% for β-carotene/linoleic acid and DPPH tests, respectively. This is due to the fact that TPC is higher in this treatment than in the other treatments, as shown in Table 3. Additionally, the inhibition rates for ultrasound-assisted and solvent extraction were 72.58 ± 0.94% and 67.24 ± 1.05%, respectively. Our results were consistent with Arya et al. [23], which indicate that maximum inhibition was observed for the β-carotene/linoleic acid and DPPH tests when the total phenolic compounds increased due to differences in raw material and extraction methods.

### 2.7. Evaluation of Sunflower Oil after Frying

Figure 6A shows the changes in fatty acid (mg KOH/g oil) value after 15 hours of frying for 3 days. An increase in all treated samples was observed, and the order was as follows (from lowest to highest): sunflower oil fortified with SCF-CO_2_ Nano-capsules (% of the increase was 6.6%), sunflower fortified with TBHQ (% of the increase was 44.6%), and sunflower oil (% of the increase was 121.5%), respectively. In addition, the same order was observed in peroxide values (Figure 6B), in which the increase was 10.6%, 19.3%, and 36.5% in sunflower oil fortified with SCF-CO_2_ Nano-capsules, sunflower fortified with TBHQ, and sunflower oil, respectively (Figure 6A). Thus, when compared to TBHQ, the SCF-CO_2_ extract has the highest antioxidant activity, and, according to studies, supercritical fluid extraction-obtained extracts maintain or surpass the bioactivity of conventionally-obtained extracts. This is due to the specificity of the supercritical fluid extraction method, which promotes a selective extraction and yield extracts rich in desirable compounds, free of organic solvents, and unaffected by degradation or reaction [24]. In this respect, Chammem et al. [25] reported that, by adding rosemary natural extract to frying oil and comparing it to the same frying oil without antioxidants, it was discovered that the addition of rosemary natural extract improved the characteristics of the frying oil by reducing the peroxide value and preserving a greater amount of unsaturated fatty acids.

## 3. Materials and Methods

### 3.1. Materials

All chemicals and reagents used in this study were purchased from Sigma Chemical Co., Ltd. (St. Louis, MO, USA). Flowers’ petals at 40 °C in the oven were dried (Shel-lab, Suite G Naperville, IL, USA) until constant weight (48 h) and then subjected to grinding by an analytical mill to a size of 1 mm (Cole-Parmer, Vernon Hills, IL, USA), sieved up to 50 mesh and stored at refrigerator (5 °C) until further analysis. Sunflower oil with no added antioxidants was obtained from ARMA company, 10th of Ramadan City, Egypt.

### 3.2. Solvent Extraction

The *Cassia javanica* Petals Powder (CJDP) was extracted at room temperature using food-grade ethanol at different concentrations (0–90%). About 2 g of the powdered sample were mixed with 50 mL of food-grade ethanol and shaken for 24 h in dark glass bottles. The solution was then filtered with Whatman filter paper (No. 1). Finally, the filtrate was kept at −18 °C for the total phenolic determination and also for other analyses [26].

### 3.3. Ultrasound-Assisted Extraction

About 4 g of each sample were mixed with 100 mL of distilled water in a 150 mL glass beaker. A 750 W ultrasonic processor (Sonics & Materials, Inc. VCX750 Model, Newtown, CT, USA) with a 0.5-inch probe and 20 kHz frequency was used. The treatment of sonication was performed at 25 °C by applying power ranged from 100–400 W for a holding time of 5–15 min maintaining 5 s pulse durations. In order to prevent the samples from overheating, freezing water was circulated out of the treatment chamber [26].

### 3.4. Supercritical Fluid Carbon Dioxide (SCF-CO_2_) Co-Solvent Extraction

Experiments were conducted with a laboratory-scale apparatus (Speed TM SFE-2/4, Applied separations, built in conjunction with the USDA1-USA) according to Paviani et al. [27]. At a constant flow rate of 1 g/min, the CO_2_ was injected into the extraction tank, which was supported by two 300-mesh wire discs at both ends. In each experiment, approximately 0.270 kg of SCF-CO_2_ was employed and 30 min of static contact between the sample and the supercritical solvent were used. The experiments carried out at 40 °C, at different pressures (150, 200, 250, 300, and 350 bar), were used with the addition of ethanol 98.8% as co-solvent in the proportions of 15% (*w*/*w*). The following extraction conditions were employed: flow rate 10 mL/min, static state for 60 min and dynamic state for 60 min. The global extraction yields were determined by dividing the total mass extracted by the initial mass of the CJDP sample (dry basis).

### 3.5. Experimental Design and Statistical Analysis

The phenolic components in CJDP were extracted using ultrasound. The sonication time (min) and sonication power (watt) were optimized (in order to maximize polyphenol extraction yield) via response surface methodology (RSM) utilizing a central composite response surface design derived from Design-Expert version 7.0.0. (Statease Inc., Minneapolis, MI, USA, Trial version). The selected levels of sonication time (χ1) and sonication power (χ2) that resulted in the highest polyphenol content were (5–15 min) and (100–400 watts). Response data were fitted to the second-order polynomial equation, which explained the effect of the independent variables on the response, as well as their combined effect on the response Y, and determined the link between the test variables. Using a reduced cubic model with linear, squared, and interaction variables, experimental data were fitted using the interaction term. In the response surface analysis, which is defined by the generalized second-order polynomial model, was utilized (Equation (2)):Y = ao + a1 χ^1^ + a2 χ^2^ + a3 χ^3^ + a12 χ^1^ χ^2^ + a13 χ^1^ χ^3^ + a23 χ^2^ χ^3^ + a11 χ^12^ + a22 χ^22^ + a33 χ^32^(2)
where Y (i = 1–5) is the expected response of TPC, respectively, ao the estimated regression coefficient of the fitted response at the central point of the model, a1, a2, a3 the coefficient of regression for linear effect expressions, a11, a22, a33 the quadratic effects, and a12, a13, a23 the effects of interaction. Using analysis of variance, statistical significance of the model and its various terms was assessed (ANOVA). In addition to R2, adjusted R2, and predicted R2 values, the lack of fit test was employed to assess the adequacy of created models.

### 3.6. Nano-Capsule Preparation of Cassia javanica SCF-CO_2_ Extract

Nano-emulsion was prepared by the oil-in-water emulsion method to prepare the *Cassia javanica* SCF-CO_2_ extract-chitosan nano-capsules according to Esquerdo et al. [28] with some modifications.

Firstly, one gram of *Cassia javanica* SCF-CO_2_ extract was added to 5 mL deionized water (1:5 *w*/*v* ratio), and 1% of surfactant (lecithin) was added to the mixture. The lecithin solution 1% was added to the *Cassia javanica* SCF-CO_2_ extract and then deionized water was added gradually while stirring (at a speed of 2000 rpm at room temperature). A high-speed homogenizer (Model: 400ELPC, PRO Scientific Inc., 01-02411ELPC HOMOGENIZER, Oxford, CT, USA) at 20,000 rpm for 20 min was used. An ice water bath to reduce the temperature of the mixture and form a w/o nano-emulsion. The homogenization was set to 10 °C since it was found to be the optimum homogenization temperature. After the preparation, all samples were stored at 4 °C for 24 h before the encapsulation process.

Secondly, the preparation of the nano-capsules was conducted as follows: chitosan was diluted in acetic acid solution (1% *w*/*v*) and stirred utilizing a magnetic stirrer at 2000 rpm for one hour. The solution was kept overnight in a shaking water bath. About 10 mL of chitosan solution (1:5 *v*/*v* ratio) was added gradually into the emulsion mixture of previously prepared oil and water. The mixture was homogenized using the ultrasonic water bath (Ultrasonic Cleaner MTI Corporation, Model UD150SH3.8LQ, City, US State abbrev., USA) at 30 °C for 30 min.

### 3.7. Potato Fries Deep-Fried in Sunflower Oil

The procedure of deep-fat frying was conducted as the protocol described by Sharma et al. [29]. About 50 gm of potato fries were fried in 250 mL of oil sample (control). The oil was enriched with nano-capsules of *Cassia javanica* SCF-CO_2_ extract separately in a household fryer (diameter 28 cm, depth 6 cm) at 180 °C for 120 s. Timing and temperatures for deep-frying were established based on laboratory tests conducted in advance (data not shown). A second frying operation was conducted in the same 250 mL of oil under the same conditions. After frying, samples of both unfortified and fortified oils were cooled to room temperature and stored separately in brown glass bottles overnight in preparation for additional frying. After 15 h, 20 mL of oil sample were taken from control, fortified oil with *Cassia javanica* SCF-CO_2_ extract, and oil with TBHQ for Acid Value (AV) and Peroxide Value (PV) determination. The same frying processes were repeated three times with each oil sample after successive storage of oils overnight and withdrawal of a 20 mL oil sample. The total time for an experiment was 3 days.

### 3.8. Phenolic Compounds of Cassia javanica Petals Powder

According to Abedelmaksoud et al. [20,30], total phenolic compounds (TPC) of CJDP extract were determined using the Folin–Ciocalteu assay and expressed as mg Gallic acid equivalents (GAE) per gram of powder. Phenolic compounds in CJDP samples were analyzed using HPLC following the protocol described by Elsayed et al. [31]. Briefly, the petals of *Cassia javanica* (CJDP) were analyzed with an Agilent 1260 series HPLC system (Agilent Technologies Inc., Santa Clara, CA, USA). The C18 column (100 mm 4.6 mm i.d., 5 µm) was used to achieve separation. At a flow rate of 0.6 mL/min, the mobile phase contained (A) water containing 0.2% H3PO4, (B) methanol, and (C) acetonitrile. The elution gradient was as follows: 0–11 min (96 percent A, 2 percent B); 11–13 min (50 percent A, 25 percent B); 13–17 min (40 percent A, 30 percent B); 17–20.5 min (50 percent B, 50 percent C); and 20.5–30 min (96 percent A, 2 percent B). The detection wavelength (UV detector) was set to 284 nm. The injection volume was 20 µL, and the temperature of the column was maintained at 30 °C. The identity of compounds was determined by comparing their retention time to that of authentic standards. Evaluation of the compound amounts by utilizing Calibration curves.

### 3.9. Total Flavonoid Content

Total flavonoid content was determined based on a colorimetric method described by Moreno et al. [32]. The results were expressed as grams of quercetin equivalents per 100 g of dry extract. The concentration of flavonoids was calculated according to the following Equation (3) obtained from the standard quercetin (20–100 µg) curve:Absorbance = 0.00001 × Quercetin in µg + 0.011 (R2 = 0.994)(3)

### 3.10. β-Carotene/linoleic Acid Assay

By measuring the inhibition of volatile organic compounds and conjugated diene hydroperoxides resulting from linoleic acid oxidation, antioxidant capacity was determined. A stock solution of a mixture of β-carotene and linoleic acid was prepared as follows: In 1 mL of HPLC-grade chloroform, 0.5 mg β-carotene was dissolved, and 25 μL linoleic acid and 200 mg Tween 40 were added. Using a vacuum evaporator, chloroform was completely evaporated. Then, 100 mL of oxygen-saturated (30 min 100 mL/min) distilled water were added with vigorous shaking; 2.5 mL of this reaction mixture were dispersed into test tubes, 350 μL portions of the extracts prepared at 2 g/L concentrations were added, and the emulsion system was incubated at room temperature for up to 48 h. The same procedure was repeated using TBHQ as the positive control and a blank. After this period of incubation, the absorbance of the mixtures at 490 nm was measured. The antioxidative capacities of the extracts were compared to those of TBHQ and a blank [33].

### 3.11. DPPH Assay

The antioxidant activity of samples was determined according to the Altemimi et al. [34] method of DPPH (2, 2-diphenyl-1-picrylhydrazyl) and estimated based on the next Equation (4):Antioxidant activity% = [(absblank − abssample)/absblank] × 100(4)
where absblank and abssample refer to the absorbance of blank and sample, respectively. BHT was used as a positive control.

### 3.12. Cytotoxic Effect on Human Cell Lines

Mitochondrial-dependent reduction of yellow MTT (3-(4,5-dimethylthiazol-2-yl)-2,5-diphenyl tetrazolium bromide) to purple formazan was used to determine cell viability [35]. All procedures were performed in a sterile environment using a biosafety class II Laminar flow cabinet (Baker, SG403INT, Sanford, ME, USA). Cells were suspended in DMEM-F12 medium [(for HePG2, MCF7, PACA2 and HCT116)] alongside a normal cell line (BJ1), 1% antibiotic-antimycotic mixture (10,000 U/mL Potassium Penicillin, 10,000 µg/mL Streptomycin Sulfate and 25 µg/ mL Amphotericin B) and 1% L-glutamine at 37 °C under 5% CO_2_. Cells were cultured in batches for 10 days, then seeded at a concentration of 10 × 10^3^ cells/well in fresh complete growth medium in 96-well microtiter plastic plates at 37 °C for 24 h under 5% CO_2_ using a water-jacketed Carbon dioxide incubator (Sheldon, TC2323, Cornelius, OR, USA). Media was aspirated, fresh medium (without serum) was added, and cells were incubated either alone (negative control) or with different concentrations of sample to give a final concentration of (100-50-25-12.5-6.25-3.125-0.78 and 1.56 ug/mL). After 48 h of incubation, the medium was aspirated, 40 µL MTT salt (2.5 μg/mL) were added to each well and incubated for a further four hours at 37 °C under 5% CO_2_. To stop the reaction and dissolve the formed crystals, 200 μL of 10% Sodium dodecyl sulfate (SDS) in deionized water were added to each well and incubated overnight at 37 °C. A positive control composed of 100 µg/mL was used as a known cytotoxic natural agent that gives 100% lethality under the same conditions [36,37]. The absorbance was then measured using a microplate multi-well reader (Bio-Rad Laboratories Inc., model 3350, Hercules, CA, USA) at 595 nm and 620 nm as a reference. The statistical significance between samples and negative control (vehicle-containing cells) was evaluated using an independent SPSS 11 *t*-test. The solvent used to dissolve plant extracts was DMSO, and its final concentration on the cells was less than 0.2%. According to the following formula, the percentage of change in viability was calculated: ((Reading of extract/Reading of negative control) − 1) × 100. Using the SPSS 11 program, the IC_50_ and IC_90_ were determined using probit analysis. The degree of selectivity of the synthetic compounds was expressed as SI = IC_50_ of the pure compound in a normal cell line/IC_50_ of the same pure compound in a cancer cell line, where IC_50_ is the concentration required to kill 50 percent of a cell population.

### 3.13. Evaluation of Sunflower Oil

To estimate the frying stability of sunflower oil as control (without fortification) in comparison with two oil samples fortified with a *Cassia javanica* SCF-CO_2_ extract nano-capsule, and TBHQ is done by estimating the stability of sunflower oil during frying by peroxide value (PV) and free fatty acid (FFA).

#### 3.13.1. Acid Value (AV) of Sunflower Oil

After each frying cycle, samples of oil were analyzed for free fatty acids (FFA) via titration in accordance with the PORIM test method p2.5 [38]. Express the result in the following Equation (5):The ratio of essential fatty acids as oleic acid = (28.2 × N × V) / W(5)
where: N = normality of NaOH solution; V = volume of NaOH solution used in mL; and W = weight of the sunflower oil sample.

#### 3.13.2. Peroxide Value (PV) of the Sunflower Oil

Changes in the peroxide value of all examined oil samples fortified with nano-forms of *Cassia javanica* SCF-CO_2_ extract were measured according to AOAC [39]. PV (meq/kg) was estimated as follows:PV (meq/kg) = (C × (V − V0) × 12.69 × 78.8)/m (6)
where C is the sodium thiosulphate concentration (mol/L), V and VO represent the volumes of sodium thiosulphate exhausted by the oil fortified and control respectively (mL), and m is the mass of cracker sample extracts (mg).

### 3.14. Statistical Analysis

The statistical analysis software used during this study was SPSS (SPSS., Chicago, IL, USA), for which the experiments were conducted in triplicate. To compare means, Duncan’s multiple comparison tests were used (*p* < 0.05) and were considered statistically significant.

## 4. Conclusions

The recovery of phenolic compounds from *Cassia javanica* L. petals was improved well by SCF-CO_2_ co-solvent and ultrasound compared to ethanolic extraction. In particular, TPC in *Cassia javanica* L. petals extract was greatly enhanced by optimum conditions. Under the optimal conditions obtained by screening as for ethanolic and SCF-CO_2_ and by RSM for ultrasound treatment, the TPCs in *Cassia javanica* L. petal extracts by ethanolic, SCF-CO_2_, and ultrasound were up to 92.2 mg/g, 177.58 mg/g, and 112.89 mg/g, respectively. Additionally, based on the β-carotene/linoleic acid and DPPH free radical-scavenging test systems, the antioxidant activity of *Cassia javanica* L. petals extract by SCF-CO_2_ extraction was significantly stronger than that of the ultrasound, ethanolic extract, and also BHT (*p* < 0.05). SCF-CO_2_ co-solvent recorded the highest inhibition % for PC3 and MCF7 and the lowest IC50 values for PC3 and MCF7. Finally, the use of supercritical CO_2_ as a green extraction method resulted in the highest concentration of TPC, which correlated to antioxidant activity, anticancer activity, and sunflower oil’s oxidative stability. The study was conducted on a laboratory scale and will need to be optimized for industrial-scale application. Further studies on the antimicrobial activity of *Cassia javanica* L. petals extract need to be conducted.

## Figures and Tables

**Figure 1 molecules-27-04329-f001:**
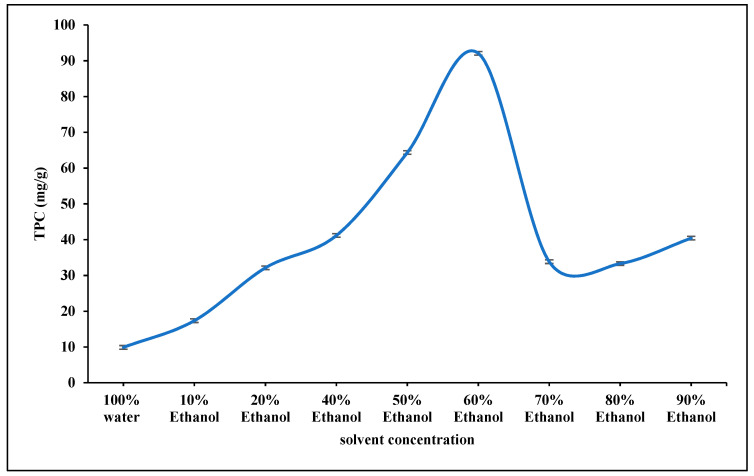
Effect of ethanol% on the TPC extraction from CJPD.

**Figure 2 molecules-27-04329-f002:**
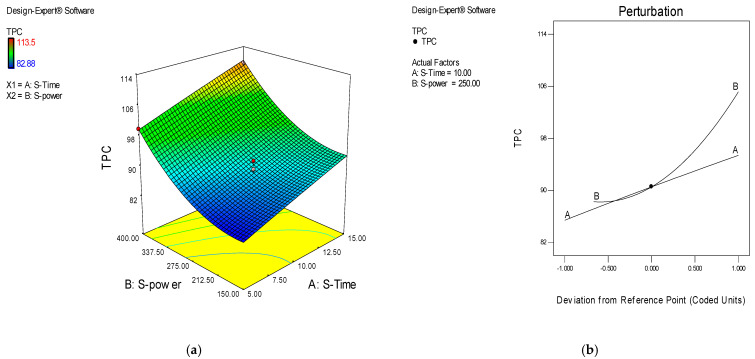
(**a**) Effect of ultrasound parameters (sonication time (factor A) and sonication power (factor B)) on the TPC–response surface and contour plots; (**b**) perturbation plot showing the relative significance of factors on the TPC values.

**Figure 3 molecules-27-04329-f003:**
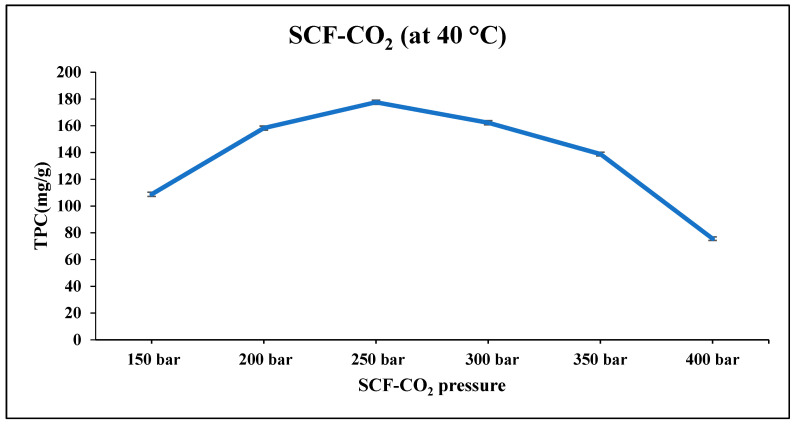
Effect of SCF-CO_2_ co-solvent pressure on TPC of CJPD.

**Figure 4 molecules-27-04329-f004:**
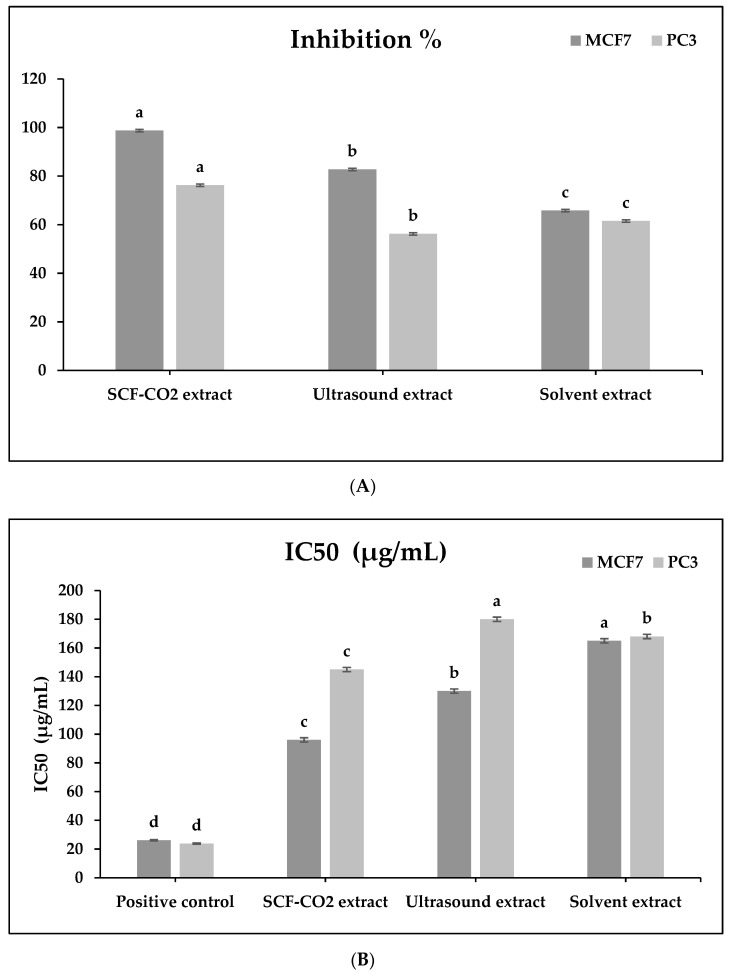
(**A**) and (**B**) Anticancer effect of *Cassia javanica* petals extract treated by ethanol, ultrasound, and SCF-CO_2_. IC_50_: Lethal concentration of the sample which causes the death of 50% of cells in 48 h; PC3: Prostate cell line; MCF7: Human Caucasian breast adenocarcinoma, Positive control Adriamycin (Doxorubicin). a–d different superscripts in a column were significantly different (*p* < 0.05); data are means ± S.D. (*n* = 3).

**Figure 5 molecules-27-04329-f005:**
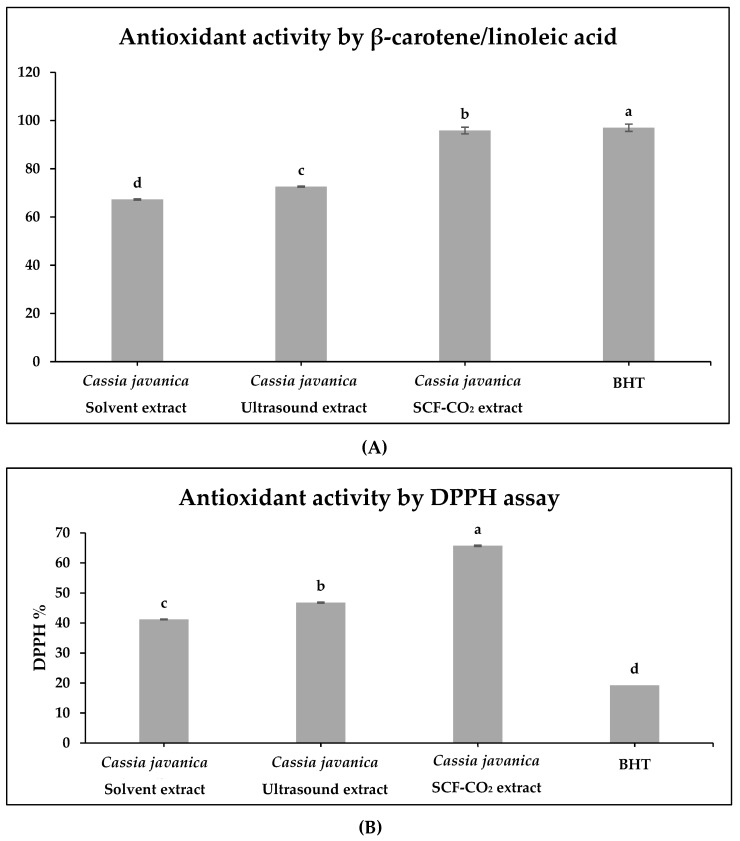
Antioxidant activity by (**A**) β-carotene/linoleic acid assay (inhibition ratio %); (**B**) DPPH assay of CJDP treated samples. BHT was used as a positive control. a–d different superscripts in a column were significantly different (*p* < 0.05); data are means ± S.D. (*n* = 3).

**Figure 6 molecules-27-04329-f006:**
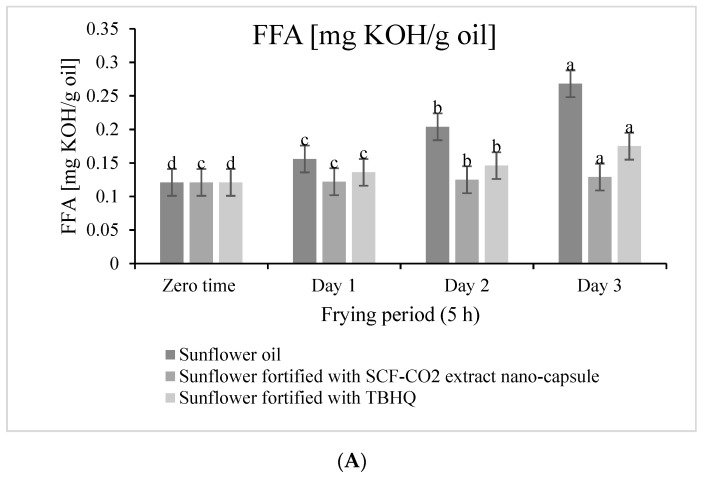
Changes of FFA (**A**) and PV (**B**) of fortified sunflower oil compared with control samples. I %: Relative Increase % of FFA or PV after 15 h of frying at 3 days; a-d different superscripts in a column were significantly different (*p* < 0.05) values are mean ± S.D. for triplicate determinations.

**Table 1 molecules-27-04329-t001:** The experimental design used and values of the response.

Run	S-Time * (min)	S-Power * (watt)	TPC (mg/g)
1	3	250	83.05 ± 0.22
2	10	250	90.25 ± 0.15
3	5	150	83.83 ± 0.18
4	17	250	98.42 ± 0.12
5	5	400	99.95 ± 0.15
6	10	250	92.56 ± 0.20
7	10	150	88.36 ± 0.11
8	15	150	92.08 ± 0.22
9	3	250	82.88 ± 0.25
10	10	250	90.32 ± 0.28
11	15	400	109.24 ± 0.21
12	10	250	88.9 ± 0.13
13	10	450	113.5 ± 0.24

* S-Time: Sonication time; S-power: Sonication power.

**Table 2 molecules-27-04329-t002:** Regression coefficients of the fitted second-order polynomials representing the relationship between the responses and variables.

Source	Squares	Df	Square	Value	Prob > F	
Model	1089.05	5	217.81	141.26	<0.0001	significant
A-S-Time	228.20	1	228.20	148.00	<0.0001	
B-S-power	162.50	1	162.50	105.39	<0.0001	
AB	0.074	1	0.074	0.048	0.8332	
A2	0.14	1	0.14	0.092	0.7703	
B2	85.71	1	85.71	55.59	0.0001	
Residual	10.79	7	1.54			
Lack of Fit	3.88	3	1.29	0.75	0.5773	Not significant
Pure Error	6.91	4	1.73			
Cor Total	1099.85	12				
Std. Dev.	1.24	R^2^	0.9902			
Mean	93.33	Adj R^2^	0.9832			
C.V. %	1.33	Pred R^2^	0.9606			
PRESS	43.32	Adeq Precision	35.282			

**Table 3 molecules-27-04329-t003:** Predicted and actual values of TPC and TFC treated by the optimum ultrasound conditions.

Treatment	TPC (mg/g)		TFC (mg QE/g)
	Predicted Value	Actual Value	
Ethanol 60%	-	92.1	67.43
Sonication (15 min, 400 watt)	110.147	112.89	82.54
SCF-CO_2_	-	177.58	139.85

**Table 4 molecules-27-04329-t004:** Total phenolic compounds and identification of phenolic compounds.

Compounds	Solvent Extraction (mg/kg)	Ultrasound (mg/kg)	SCF-CO_2_ (mg/kg)
Pyrogallol	25.23 ± 0.05 b	16.44 ± 0.04 c	80.19 ± 0.06 a
Gallic acid	-	19.82 ± 0.03 a	-
Catechol	143.72 ± 0.07 b	227.16 ± 0.05 a	-
*p*-Hydroxy benzoic acid	66.63 ± 0.09 c	124.96 ± 0.07 b	1981.65 ± 0.04 a
Chlorogenic	1.91 ± 0.03 c	4.95 ± 0.05 b	45.81 ± 0.07 a
Vanillic acid	13.48 ± 0.02 c	53.95 ± 0.02 b	314.39 ± 0.06 a
Syringic acid	10.08 ± 0.05 c	26.57 ± 0.03 b	423.23 ± 0.04 a
Benzoic acid	297.22 ± 0.01 c	383.09 ± 0.06 b	5478.37 ± 0.05 a
Ferulic acid	6.14 ± 0.08 c	8.20 ± 0.05 b	20.79 ± 0.06 a
Rutin	238.11 ± 0.02 c	497.16 ± 0.04 b	4891.56 ± 0.05 a
Ellagic	32.14 ± 0.03 a	29.56 ± 0.08 b	-
o-Coumaric acid	-	-	18.17 ± 0.07 a
Cinnamic acid	-	-	9.61 ± 0.05 a
Quercitin	1315.75 ± 0.04 c	2980.92 ± 0.05 b	50,018.10 ± 0.08 a
Rosemarinic	202.43 ± 0.03 b	656.39 ± 0.04 a	-
Myricetin	13.78 ± 0.02 c	19.67 ± 0.02 b	421.88 ± 0.05 a
Kampherol	23.62 ± 0.01 b	2.45 ± 0.07 c	47.51 ± 0.02 a

Statistically significant difference shown levels a, b, c compared with same column (*p* ≤ 0.05); data are means ± S.D. (*n* = 3).

## Data Availability

Not applicable.

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
