# Peer review of "Optimized Green Extraction of Polyphenols from Cassia javanica L. Petals for Their Application in Sunflower Oil: Anticancer and Antioxidant Properties"

_molecules, 2022, doi:10.3390/molecules27144329_

Round 1

Reviewer 1 Report

The manuscript molecules-1778336, “Optimized green extraction…”by M. I. Younis at. al., is an interesting material that, in my opinion, can be published in “Molecules” after some clarifications and improvements.  The main concern is related to the procedure by which the results are reported.  For each type of result, the authors indicate bibliographic references (e.g. ref. 9, 10, 11, 16).  It is not clear if the authors made the measurements using the indicated procedures from the literature, or they only used the data from those references.  In case that no measurement was done in their own study, the uniformity of the results is highly questionable.  In case the results are original, this should be stated and a detailed description of the analysis procedure must be included.    Also, the English needs some improvement.   In the introduction, for example, sentence such “Because plant-based products have no adverse effects and are relatively inexpensive” must be clarified.  Also, in introduction, note that active compounds in the plants were started to be identified rather long time ago (see e.g. D. Jeffreys, “Aspirin: The remarkable story of a wonder drag”, Bloomsbury Pub. New York, 2005)   

Author Response

Reviewer 1

 The main concern is related to the procedure by which the results are reported.

For each type of result, the authors indicate bibliographic references (e.g. ref. 9, 10, 11, 16).

It is not clear if the authors made the measurements using the indicated procedures from the literature, or they only used the data from those references.

In case no measurement was done in their own study, the uniformity of the results is highly questionable.

In case the results are original, this should be stated and a detailed description of the analysis procedure must be included.

Response:

Yes, this is our original data in which we have conducted screening for ethanol concentration for total phenolic extraction as in figure 1 and the procedure is already included in the material and method section (3.2. Solvent Extraction).  Regarding ref. 9, 10, and 11 just we have mentioned them to clarify the extraction of TPC in other plant materials that comply with our results regarding the improvement the extraction of TPC with increasing the ethanol concentration ranged from 50 to 75% (please see lines – in the revised manuscript).

Regarding ref. 16, we have removed it and made a good rewriting for section 2.4 in the results and discussion part (please see lines – in the revised manuscript).

Also, the English need some improvement.

Response: Done we have made a good revision through the manuscript.

In the introduction, for example, a sentence such as “Because plant-based products have no adverse effects and are relatively inexpensive” must be clarified.

Also, in the introduction, note that active compounds in the plants were started to be identified a rather long time ago (see e.g., D. Jeffreys, “Aspirin: The remarkable story of a wonder drag”, Bloomsbury Pub. New York, 2005)

Response: done, we have rewritten it to be “Natural compounds in plants for maintaining human health started to be identified a rather long time ago, though the identification of active compounds from this valuable source came to light in recent years. The potential adverse effects of plant-based products are very limited in addition to relatively inexpensive compared to chemical substances ( Jeffreys, 2008)”.

Reviewer 2 Report

Authors describes the Optimized green extraction of polyphenols from Cassia javanica L. petals for its application in sunflower oil: anticancer and antioxidant properties.

The work has numerous criticalities that make it unsuitable for publication. 

Firstly the antioxindat activity ewas evaluated by just one test. I multi target approach is necessary to evaluate the antioxidant activity (See a Review article of Antolovich et al. )

More importantly, which polyphenols resist the temperatures of use of this oil?

The experimental design lacks logical sense.

TPCs are determined spectrophotomentrically with all the limitations of this technique. Why weren't the TFCs determined too?

Why these cell lines? What does the anti-cancer activity have to do with it?

A positive control is strickly necessary.

Conclusion and abstract are very confused.

Discussion is lacking of content.

In my opinion this manuscript is not suitable for publication in this journal.

Author Response

The authors describe the Optimized green extraction of polyphenols from Cassia Javanica L. petals for its application in sunflower oil: anticancer and antioxidant properties. The work has numerous criticalities that make it unsuitable for publication.

Firstly, the antioxidant activity was evaluated by just one test. I multi-target approach is necessary to evaluate the antioxidant activity (See a Review article of Antolovich et al.)

Response: we have conducted and included the DPPH assay please see figure 5 B in the revised manuscript.

More importantly, which polyphenols resist the temperatures of use of this oil?

Response: The main objective of this study was to investigate the effect of all treatments on the TPC then the best condition for each treatment was subjected to fractionation of phenolic compounds (Table 4). After that, we have made a Nano encapsulation for the best treatment (SCF-CO2) and applied it in oil to investigate the effect of TPC on the oil quality.

The experimental design lacks logical sense.

Response: For ethanolic and SCF-CO2 we have made a screening for select the best condition for TPC extraction which there is no more than one valuable factor to make RSM and also based on the literature we have selected the rage of selected factor for our experiments. while in sonication we have factors as we selected to optimize the conditions for TPC extraction by RSM.

TPCs are determined spectrophotometrically with all the limitations of this technique. Why weren't the TFCs determined too?

Response: we have determined Total flavonoids and the results were included in table 3.

Why these cell lines? What does the anti-cancer activity have to do with it?

Response: we have found many studies that used these cell lines to examine different plant extracts and this is also very prevalent in Egypt and the world. Thus, the results were very encouraging for their selection based on this. On the other hand, we were too afraid to work on other cell lines that were resistant to our extract, and thus maybe it will not give us good cytotoxicity results.

  • Gaidhani, S. N., Lavekar, G. S., Juvekar, A. S., Sen, S., Singh, A., & Kumari, S. (2009). In-vitro anticancer activity of standard extracts used in ayurveda. Pharmacognosy Magazine5(20), 425.
  • Shoemaker, M., Hamilton, B., Dairkee, S. H., Cohen, I., & Campbell, M. J. (2005). In vitro anticancer activity of twelve Chinese medicinal herbs. Phytotherapy Research: An International Journal Devoted to Pharmacological and Toxicological Evaluation of Natural Product Derivatives19(7), 649-651.
  • Solowey, E., Lichtenstein, M., Sallon, S., Paavilainen, H., Solowey, E., & Lorberboum-Galski, H. (2014). Evaluating medicinal plants for anticancer activity. The Scientific World Journal2014.
  • Rehana, D., Mahendiran, D., Kumar, R. S., & Rahiman, A. K. (2017). Evaluation of antioxidant and anticancer activity of copper oxide nanoparticles synthesized using medicinally important plant extracts. Biomedicine & Pharmacotherapy89, 1067-1077.
  • Mykhailenko, O., Lesyk, R., Finiuk, N., Stoika, R., Yushchenko, T., Ocheretniuk, A., ... & Georgiyants, V. (2020). In vitro anticancer activity screening of Iridaceae plant extracts. Journal of Applied Pharmaceutical Science10(7), 059-063.

A positive control is strictly necessary.

Response: yes, this is very necessary point so we have included positive control in figure 4A. Also, regarding antioxidant activity: BHT positive control was added to figure 5.

The conclusion and abstract are very confused.

Response: we have made a good revision and rewritten the conclusion and abstract in the revised manuscript.

Discussion is lacking in content.

Response: we have added and made a good clarification for discussion section in the revised manuscript.

Reviewer 3 Report

Manuscript ID: molecules-1778336

In the article entitled: “Optimized green extraction of polyphenols from Cassia javanica L. petals for its application in sunflower oil: anticancer and antioxidant properties” was study phenolic compounds from Cassia javanica petals powder (CJDP) using a variety of extraction methods, including solvent extraction, ultrasound-assisted extraction, and SCF-CO2 extraction. Also analyzed the activities of these extracted compounds before applying them to sunflower oil without added antioxidants.

 Title

The title and the aim of the study are clearly constructed.

Abstract

The abstract includes the aim of the study, methods used in the experiment and contain the principal results and conclusions.

Introduction

The introduction describes the matter of the experiment accurately and clearly states the problem being investigated.

Methods

The data is well collected. The methods are described in detail, in the way which permits the research to be replicated. The sampling is appropriate and adequately described.

Results

The results were discussed extensively, in a clear and legible way.

Discussion

They correctly interpreted and described the significance of the results for the research. They skillfully referred to the results of other researchers.

References

The references are accurate.

Language

The article is correctly written.

Author Response

Response: Thank you very much for your time and clarifying the strong points and the advantages of our manuscript.

Reviewer 4 Report

In the present study, the others explore the extract phenolic compounds from Cassia javanica petals powder using various extraction methods to investigate the activities such as anticancer and antioxidant of these extracted compounds before applying them to sunflower oil without added antioxidants. This is a very interesting paper. Furthermore, this work in present form presents some imperfections according to the following comments.

Comments to Authors:

1-         Page 1; lines 41-43: This information requires one or more appropriate bibliographic references.

 2- Page 2; Standardized the writing of the abbreviation of “Cassia javanica Petals Powder” in all the text

Like:

-           Line 79: Cassia javanica petals powder (CJDP)

-           Line 86: Cassia javanica Petals Powder (CJPD)

3- Page 5; line 161: The sentence begins with the capital letter.

4- Page 6; figure 3: Appropriately modify the abscissa axis.

5- Page 8; Figure 4A: Add standard deviation.

6- Page 9; Figure 5: Write in italics “Cassia Javanica” in the abscissa axis. To be able to compare the antioxidant activity, add the positive control.

7- Page 10; figure 6A: I notice that your standard deviations are quite high. do you have an explanation?

8- Page 10; line 326: "Flowers’ petals at 40°C in the oven were dried (Shel-lab, USA)”. Why the choice of this drying and not that at room temperature in the dark to avoid routes of degradation of biomolecules by heat.

9- Page 10; line 328: "….. sieved up to 50 mesh and stored at refrigerator (5°C)”. If I understand correctly you store Flowers' petals powder at 5°C, for how times?

10- Standardized the writing of the units’ ul, ml by uL, mL (like page 13 section “Cytotoxic effect on human cell lines”) in all the text.

11- Page 13; lines 470 and 472 correct the writing of IC90 and IC50 (in subscript). The same note for the writing of CO2 and some chemical compounds in all paper.

12- Page 14; line 483: The ratio of essential fatty acids as oleic acid = (28.2 × N × V) / W.........

The dots correspond to what?

Author Response

In the present study, the others explore the extract phenolic compounds from Cassia javanica petals powder using various extraction methods to investigate the activities such as anticancer and antioxidant of these extracted compounds before applying them to sunflower oil without added antioxidants. This is a very interesting paper. Furthermore, this work in present form presents some imperfections according to the following comments.

Comments to Authors:

  • Page 1; lines 41-43: This information requires one or more appropriate bibliographic references.

Response: The bibliographic was added to lines mentioned above

  • Page 2; Standardized the writing of the abbreviation of “Cassia javanica Petals Powder” in all the text

Like:

- Line 79: Cassia javanica petals powder (CJDP)

- Line 86: Cassia javanica Petals Powder (CJPD)

Response: Standardization was done for all the paper

  • Page 5; line 161: The sentence begins with the capital letter.

Response: Done

  • Page 6; figure 3: Appropriately modify the abscissa axis.

Response: Done

  • Page 8; Figure 4A: Add standard deviation.

Response: Done

  • Page 9; Figure 5: Write in italics “Cassia Javanica” in the abscissa axis. To be able to compare the antioxidant activity, add the positive control.

Response: All “Cassia Javanica” are now written in italics and BHT positive control was added for comparison.

  • Page 10; figure 6A: I notice that your standard deviations are quite high. do you have an explanation?

Response: It was a mistake while adjusting the error bar.

  • Page 10; line 326: "Flowers’ petals at 40°C in the oven were dried (Shel-lab, USA)”. Why the choice of this drying and not that at room temperature in the dark to avoid routes of degradation of biomolecules by heat.

Response:

  • Because it was intended for industrial use and drying at room temperature takes time.
  • Most of the studies used temperatures less than 60 Celsius as a drying temperature for plants that are considered a source of polyphenols to minimize losses in polyphenols and no significant change was observed. (Larrauri, J. A., Rupérez, P., & Saura-Calixto, F. (1997). Effect of drying temperature on the stability of polyphenols and antioxidant activity of red grape pomace peels. Journal of agricultural and food chemistry, 45(4), 1390-1393.)

  • Page 10; line 328: "….. sieved up to 50 mesh and stored at refrigerator (5°C)”. If I understand correctly, you store Flowers' petals powder at 5°C, for how times?

Response: The powder was stored at 5°C for 18 hours before first analysis and then stored in the freezer at -18°C.

  • Standardized the writing of the units’ ul, ml by uL, mL (like page 13 section “Cytotoxic effect on human cell lines”) in all the text.

Response: Done

  • Page 13; lines 470 and 472 correct the writing of IC90 and IC50 (in subscript). The same note for the writing of CO2 and some chemical compounds in all paper.

Response: Done

12- Page 14; line 483: The ratio of essential fatty acids as oleic acid = (28.2 × N × V) /W.........

The dots correspond to what?

Response: it was a typing mistake, so we removed them.

Round 2

Reviewer 2 Report

Authors sufficiently improved the manuscript.

Reviewer 4 Report

The authors have made all the necessary corrections and answered all the questions, I recommend the publication of this paper.